# Non-universal transmission phase behaviour of a large quantum dot

Hermann Edlbauer[1], Shintaro Takada[1,2], Grégoire Roussely[1], Michihisa Yamamoto[3,4], Seigo Tarucha[3,4], Arne Ludwig[5], Andreas D. Wieck[5], Tristan Meunier[1] & Christopher Bäuerle [1]

The electron wave function experiences a phase modification at coherent transmission through a quantum dot. This transmission phase undergoes a characteristic shift of $\pi$ when scanning through a Coulomb blockade resonance. Between successive resonances either a transmission phase lapse of $\pi$ or a phase plateau is theoretically expected to occur depending on the parity of quantum dot states. Despite considerable experimental effort, this transmission phase behaviour has remained elusive for a large quantum dot. Here we report on transmission phase measurements across such a large quantum dot hosting hundreds of electrons. Scanning the transmission phase along 14 successive resonances with an original two-path interferometer, we observe both phase lapses and plateaus. We demonstrate that quantum dot deformation alters the sequence of phase lapses and plateaus via parity modifications of the involved quantum dot states. Our findings set a milestone towards an comprehensive understanding of the transmission phase of quantum dots.

[1] Univ. Grenoble Alpes, CNRS, Grenoble INP, Institut Néel, 38000 Grenoble, France. [2] National Institute of Advanced Industrial Science and Technology (AIST), National Metrology Institute of Japan (NMIJ), 1-1-1 Umezono, Tsukuba, Ibaraki 305-8563, Japan. [3] Department of Applied Physics, University of Tokyo, 7-3-1 Hongo, Bunkyo-ku, Tokyo 113-8656, Japan. [4] RIKEN Center for Emergent Matter Science (CEMS), 2-1 Hirosawa, Wako-shi, Saitama 31-0198, Japan. [5] Lehrstuhl für Angewandte Festkörperphysik, Ruhr-Universität Bochum, Universitätsstraße 150, 44780 Bochum, Germany. H. Edlbauer and S. Takada contributed equally to this work. Correspondence and requests for materials should be addressed to C.B. (email: christopher.bauerle@neel.cnrs.fr)

The phase of the electron wave function lies at the heart of coherent transport phenomena such as universal conductance fluctuations or weak localisation[1–3]. One way of accessing this quantity employs a quantum interferometer. Interesting phenomena arise when a quantum dot (QD) is inserted into one branch of such an interferometer. Then a Coulomb blockade is present and the wave function of an electron experiences a phase modification at resonant transfer through the QD. As one scans through a Coulomb blockade peak, this transmission phase gradually changes. The magnitude of this phase shift strongly depends on the coupling of the QD to the leads in the interferometer branch. In the strong coupling regime, the transmission phase shift is determined by the electron occupancy of the highest occupied energy level. When this level only hosts a single electron, its spin plays a predominant role. Below a certain temperature threshold, in this case the electron forms a strongly correlated many body state—the Kondo ground state—, which leads respectively to a $\pi/2$ phase shift across two consecutive resonances[4,5]. For weak coupling, on the other hand, a phase shift of $\pi$ is theoretically expected to occur along a Coulomb blockade peak and has been measured for the first time in 1997[6]. The course of this shift can be understood by Friedel's sum rule and follows a Breit–Wigner profile[7].

A puzzling situation arises when scanning through several consecutive resonances. In this case, the transmission phase behaviour in between the resonances in principle depends on the spatial symmetries of the QD states[8,9]: If the involved orbitals have the same parity, a sudden phase lapse of $\pi$ appears in the valley between two consecutive resonances. When the orbital parity is changing, on the other hand, such a lapse is absent giving rise to a phase plateau. Pioneering experiments have investigated the transmission phase across a large QD hosting about 200 electrons[6]. Surprisingly, for this situation only phase lapses of $\pi$ have been found in between each of the investigated resonances. Since the measured series of phase lapses were robust against changes in various QD properties, the behaviour was termed universal. A different behaviour was observed in smaller QDs hosting only a few electrons[10]. With an electron number below 10, the occurrence of phase lapses in between the resonances was found to depend on the QD properties. Above 14 electrons in the QD, however, only phase lapses were observed giving support for a universal transmission phase behaviour[10]. Several theoretical models are devoted to explain the occurrence of phase lapses in a universal regime proposing a mechanism to make the appearance of a lapse more likely for a larger QD[11–21]. Despite these theoretical efforts, there is at present no satisfactory explanation for the complete absence of phase plateaus and the question about a universal transmission phase behaviour remains as one of the longest standing puzzles of mesoscopic physics. Transmission phase measurements are rare due to their experimental difficulty. Only a few groups have succeeded in performing such measurements[6,10,22,23] and, therefore, not much data are available to be confronted with theory.

Here we employ a recently developed Mach–Zehnder type electron interferometer[24,25] to address this long standing problem about a universal transmission phase behaviour experimentally. The main advantage of this original design is the suppression of electrons encircling the interferometer structure. It avoids multi-path contributions to the phase measurement and ensures reliable two-path interference. Taking this new path of electron interferometry, we investigate the transmission phase of a large QD having similar dimensions as in ref. [6]. Our measurements clearly show a non-universal transmission phase behaviour, where the absence of phase lapses is possible. Additionally, we find characteristic features of a parity-dependent transmission phase behaviour and demonstrate that the sequence of phase lapses and plateaus can be modified by QD deformation.

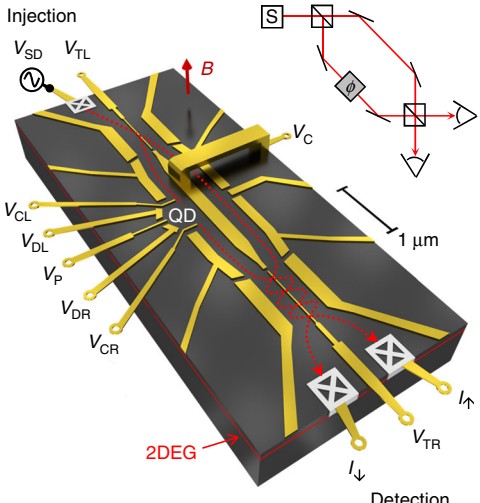

**Fig. 1** Scheme of the electron two-path interferometer. Shown is a detailed 3D view on the Schottky gates (golden) defining the conductive paths (red, dotted lines) and the interferometer structure in the two-dimensional electron gas (2DEG) located 110 nm below the surface. The ohmic contacts establishing electrical connection to the 2DEG are schematically indicated via the grey crossed boxes in the terminals of the interferometer. The inset at the upper right depicts the principle of Mach–Zehnder interferometry— the photonic counterpart of our electron interferometer

## Results

**Measurement principle**. The transmission phase measurement is based on Mach–Zehnder interferometry. Exploiting the Aharonov–Bohm (AB) effect, we measure an electronic interference pattern that is used to deduce the transmission phase of the QD. Different from the electronic Mach–Zehnder interferometer using edge states in the quantum Hall regime[26–28], it is operated at low magnetic fields. A scheme of the electron two-path interferometer is shown in Fig. 1. It is an effective three terminal device realized in an AB ring, which is sandwiched between two tunnel-coupled wires: the tunnel barrier at the injection side serves as a beam splitter guiding injected electrons into the two branches of the interferometer. A QD embedded in the lower branch of the AB ring modifies the phase of the electron wave passing through. At the detection side, a second tunnel-coupled wire guides the interfering electron waves—from the upper and lower interferometer branch—towards a pair of terminals where the currents, $I_\uparrow$ and $I_\downarrow$, are measured.

Changing the magnetic flux density, $B$, of a magnetic field perpendicular to the two-dimensional electron gas (2DEG), we observe AB oscillations in $I_\uparrow$ and $I_\downarrow$. By tuning the tunnel barriers via the voltages, $V_{TL}$ and $V_{TR}$, we obtain anti-phase AB oscillations. These anti-phase AB oscillations in $I_\uparrow$ and $I_\downarrow$ are the characteristic feature to ensure reliable two-path interference as shown by analytical quantum mechanical calculations, computer simulations and experimental investigations[24,25,29,30]. The conductance though the QD is controlled by the so-called plunger gate, P, that is embedded in the QD structure. This electrode affects the electrostatic potential of the QD and is used to bring a QD state in resonance with the leads. Figure 2a shows a Coulomb blockade peak—the electrical conductance through the QD as the plunger gate voltage, $V_P$, is swept along a resonance. For five positions along the resonance, the corresponding anti-phase AB oscillations in $I_\uparrow$ and $I_\downarrow$ are shown in Fig. 2b. The magnitude of the oscillations is linked with coherent transmission through the QD and, hence, is stronger at the centre of resonance. As $V_P$ is swept along the resonance, the anti-phase AB oscillations

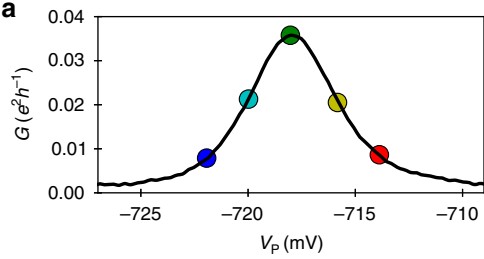

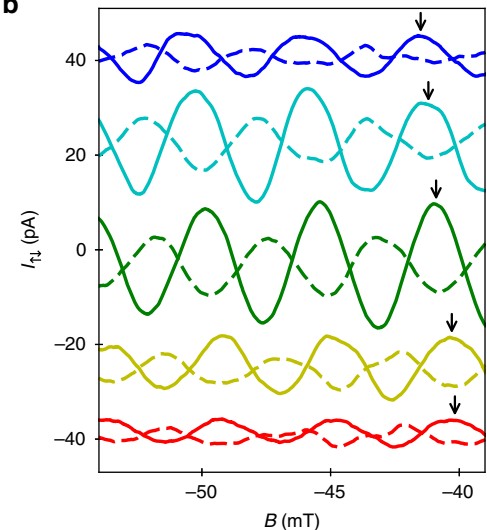

**Fig. 2** Anti-phase AB oscillations along a Coulomb blockade peak.
**a** Electrical conductance, $G$, as the plunger gate voltage, $V_P$, is swept along the resonance. **b** Excerpts of the currents, $I_\uparrow$ (solid line) and $I_\downarrow$ (dashed line), as a function of the magnetic flux density, $B$, for five positions of $V_P$ along the resonance. The $V_P$ positions of the current traces are indicated via correspondingly coloured data points in **a**. The arrows indicate the shift of the AB oscillations. For clarity, a continuous background is subtracted and an offset is added

experience a phase shift (see arrows in Fig. 2b), which directly reflects the transmission phase of the QD.

**Properties of the quantum dot**. The QD structure is defined via voltages on six Schottky gates: transversal to the transmission direction the QD is confined by the central electrode (C) and two opposing gates (DL and DR). C depletes the 2DEG below a central island to form the AB ring. The longitudinal QD confinement and the coupling to the conductive channels of the interferometer branch are defined by two narrow gates (CL and CR). The lithographic dimensions of the QD are approximately 0.5 μm in transverse direction and 0.6 μm in longitudinal direction. A scanning electron micrograph of the sample is shown in the methods section. From the electron density in the 2DEG and the size of the QD, we estimate that the QD contains about 300 electrons.

Let us now introduce further properties of the investigated QD on the basis of a Coulomb diamond measurement. The presence of the QD causes Coulomb blockade of electron transport in the lower interferometer branch allowing characterisation of the QD. By increasing the tunnel barriers via the voltages $V_{TL}$ and $V_{TR}$, electrons are steered only into the lower branch. Using lock-in detection, we then measure the transconductance, $dI_\downarrow/dV_{SD}$, and change the plunger gate voltage, $V_P$, and the DC component of the source-drain voltage, $V_{SD}$, to obtain the Coulomb diamonds. Figure 3 shows transconductance data along 14

resonances (for conductance peaks see Supplementary Fig. 1). We estimate the charging energy, $E_C$, determined by the Coulomb diamond height, as about 240 μeV[31]. From the width of the Coulomb diamonds—the spacing of the resonances—, $V_C \approx 26$ mV, we calculate the voltage to energy conversion factor as $\eta = E_C \cdot V_C^{-1} \approx 0.01\ e$. Approximating the resonances with a Lorentzian function, we obtain the coupling energy, $\Gamma \approx \eta \cdot \Delta V_{FWHM}[V] \approx 30 \sim 60$ μeV, from the full width at half maximum, $\Delta V_{FWHM}$. The energy spacing of the excited states, $\delta$, is hardly resolvable in the measured transconductance data. From the structures appearing at the resonances A11–A14, however, we estimate the level spacing as $\delta \approx 60$ μeV. Taking into account the gate geometry and the depth of the 2DEG, we can estimate the effective QD area as $A \approx 0.3$ μm × 0.4 μm. With this information, we can alternatively estimate $\delta$ from the minimum level spacing of a two-dimensional particle in a box problem[32]. We derive $\delta \gtrsim \hbar^2 \cdot \pi^2/(2 \cdot m \cdot A) \approx 50$ μeV, where $\hbar$ is the reduced Planck constant and $m$ is the effective electron mass in a GaAs crystal. The two estimates of $\delta$ are consistent and $\delta \approx \Gamma$ implies electron transport at the crossover from the single-level ($\delta > \Gamma$) to the multi-level ($\delta < \Gamma$) regime. For the investigated magnetic field range, the Zeeman energy is smaller than temperature fluctuation. We do not observe any enhancement of the valley conductance when lowering the temperature. Hence, we assume that the Kondo temperature is lower then the electron temperature.

**Transmission phase measurements**. To check the universality of transmission phase lapses in a large QD, it is necessary to investigate a set of resonances that is as large as possible. When sweeping the plunger gate over a large voltage range, however, crosstalk between the different electrostatic QD gates affects the visibility of the AB oscillations and renders the measurement more difficult. To overcome this problem, we split the total sequence of Coulomb blockade peaks into several measurements. For each, we carefully fine-tune the voltages on the electrodes defining our interferometer and QD structure to obtain maximal visibility of the anti-phase AB oscillations along the scanned resonances. In order to construct the transmission phase along the investigated resonances, the data sets have to overlap with at least one resonance.

Following this approach, we obtain the transmission phase along 14 resonances as shown in Fig. 4. Characteristic phase shifts of $\pi$ (red data points) are apparent along all of the investigated resonances labelled as A1–A14 at the corresponding peaks of conductance, $G$. Most importantly, however, the data show a significant signature of non-universal transmission phase behaviour: among the investigated 14 resonances, three times a phase lapse is absent. This leads to clear phase plateaus after the resonances A4, A10 and A13 (see black arrows in Fig. 4), where the transmission phase cumulates the characteristic shift of $\pi$. In between those plateaus, nonetheless, we also observe long sequences of phase lapses as in previous experimental investigations[6].

**Parity-dependent occurrence of phase lapses**. Theoretically, the occurrence of phase plateaus is linked to a parity change of the QD states[8,9]. For a simple one-dimensional problem, the parity should change when going from one orbital state to the next one. For a large QD, which has to be regarded as a two-dimensional system the situation is more complicated. In this case, the parity is defined by the coupling of the QD state to the two leads. One usually defines the quantity $D_n = \gamma_n^L \cdot \gamma_n^R \cdot \gamma_{n+1}^L \cdot \gamma_{n+1}^R$, where $\gamma_n^{L/R}$ is respectively the effective coupling at the left and right connection (L/R) of the $n$th QD state[8,9,17,19]. If two successive QD states, $n$ and $n+1$, have the same parity, this quantity is positive and a singularity point called transmission zero occurs that causes a phase lapse

in between the resonances. This leads to a suppression of the AB oscillation in the conductance valleys. When the parity of two successive QD states changes, on the other hand, the magnitude of the AB oscillation in the conductance valley is enhanced.

This characteristic feature of enhanced AB oscillation in conductance valleys without transmission zero is clearly apparent in our experimental data. In congruency with a parity-dependent transmission phase behaviour, we find augmented magnitude of AB oscillation, $M$, in between resonances where a phase plateau occurs (see red arrows in Fig. 4). The enhancement of the AB oscillation is strongly pronounced after the resonances A4 and A10. Our observation of this theoretically expected signature shows that transmission phase measurements indeed can be used to access the sign of the wave functions for the QD states at the connections to the leads.

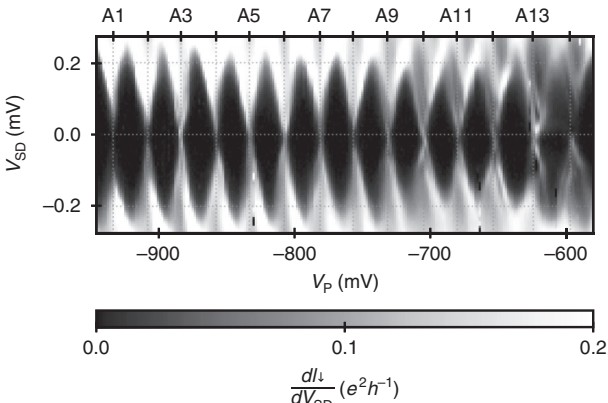

**Fig. 3** Coulomb diamonds. Transconductance measurement as the bias voltage, $V_{SD}$, and the plunger gate voltage, $V_P$, are swept. The investigated resonances, A1–A14, occurring in the swept $V_P$ interval are labelled at the upper axis

**Asymmetry in coherent transmission**. Another interesting feature that we find in the present data is the asymmetry of the AB oscillation magnitude, $M$, with respect to conductance peaks. This asymmetry around a resonance could indicate spin-flip scattering processes[33–36]. By comparing $M$ with respect to the conductance peaks, $G$, in principle the presence of an empty or partially occupied spin-degenerate level could be deduced[37,38]. If the region of reduced AB oscillation magnitude is located at the positive side of the conductance peak regarding plunger gate voltage, $V_P$, a partially occupied spin-degenerate level is present. Vice versa is the situation for an empty spin-degenerate level. Such a feature is particularly strong at resonances where phase plateaus occur. Asymmetric peaks in $M$ are also apparent at several other resonances. Nonetheless, we find no systematic correspondence to the occurrence of phase plateaus. Other effects such as the Fano effect[23,39,40] or the aforementioned transmission zero can also lead to an asymmetric AB oscillation magnitude around the resonances. The asymmetries of AB oscillation peaks and the sequence of phase plateaus point out the complexity of the internal structure of the present large QD. A detailed characterisation of the spin-state sequence for the present data is out of reach. From the occurrence of phase plateaus, however, we can clearly observe changes in orbital parity.

**Modifying the quantum dot shape**. According to a parity-dependent transmission phase behaviour[8,9], one expects modifications of the phase lapse sequence as QD states change. If, for instance, the QD shape is distorted such that the QD states and, thus, the sequence of orbital parities changes, one should observe a modified sequence of phase lapses and plateaus. With the present experimental setup, we can readily investigate this assertion. We deform the QD by changing the balance of the voltages $V_{DL}$, $V_{DR}$ and $V_P$. As a starting point for the discussion, we consider a reference QD configuration with $V_{DL} = V_{DR} = -0.92$ V. A set of transmission phase shifts measured along five consecutive resonances is shown in Fig. 5a. For this reference

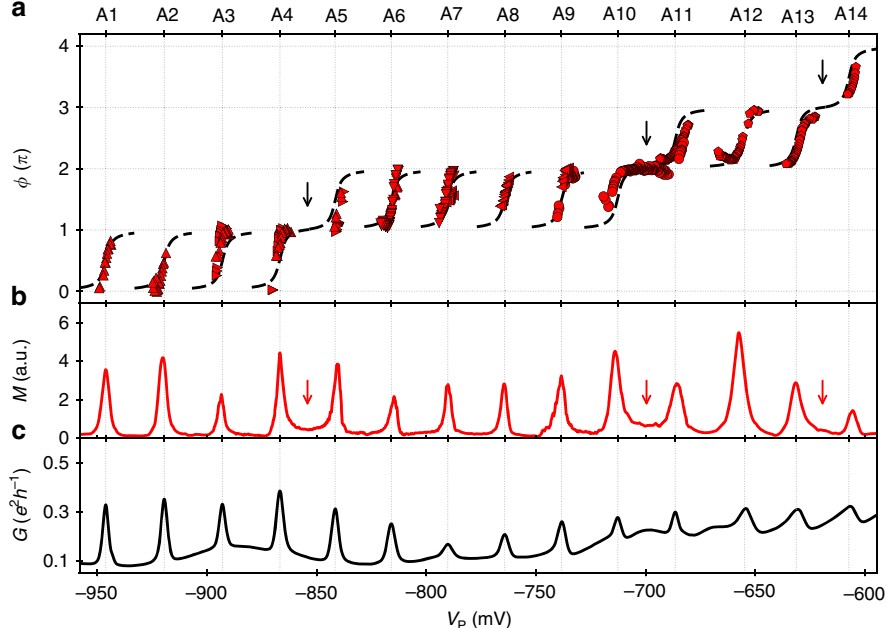

**Fig. 4** Transmission phase along 14 resonances. **a** The red data points show the transmission phase, $\phi$, measured along the resonances A1–A14. The arrows indicate phase plateaus. A guide to the eye is shown as dashed line. It is constructed from electrical conductance, $G$ (Supplementary Note 1). The data alignment procedure is described in Supplementary Note 2. **b** The magnitude of the AB oscillation, $M$, is illustrated as red line. The red arrows indicate regions of enhanced AB oscillation. **c** The electrical conductance at the lower terminal of the interferometer, $G = I_{\downarrow}/V_{SD}$, is shown as black line

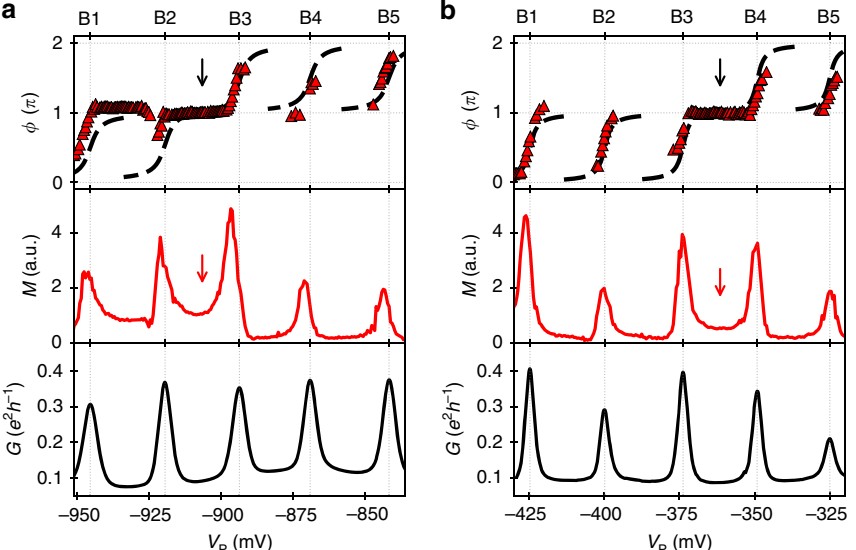

**Fig. 5** Effect of quantum dot deformation on the transmission phase. Transmission phase, $\phi$, magnitude of AB oscillation, $M$, and electrical conductance at lower terminal, $G$, along the same set of resonances (B1–B5) for different quantum dot configurations: **a** reference configuration with $V_{DL} = V_{DR} = -0.92$ V. **b** Deformed quantum dot with $V_{DL} = V_{DR} = -1.08$ V. The deviating intervals of $V_P$ in **a**, **b** result from the different gate voltage applied to the gates $V_{DL}$ and $V_{DR}$ and have been chosen such that the same resonances, B1–B5, are shown. The dashed line underneath the transmission phase data serves as guide to the eye and is constructed on the basis of conductance peak data shown in the bottom panel. The black arrows indicate the absence of phase lapses. The red arrows indicate regions of enhanced AB oscillation coinciding with phase plateaus

configuration, a phase plateau occurs after Coulomb blockade peak B2. We modify the QD shape by changing the voltages, $V_{DL}$ and $V_{DR}$, to −1.08 V. At the same time, we adjust the voltage $V_P$ to keep the electron number constant. Comparing the transmission phase measurements along the same set of resonances for the two situations—compare Fig. 5a, b—we find significant changes: We observe an alternation of the phase lapse sequence as the QD shape is modified. After deformation of the QD, the phase plateau now occurs after resonance B3 instead of B2. The deformation causes indeed a change in the sequence of orbital parities what is directly reflected in the observed course of the transmission phase. According to the altered position of the phase plateau augmented AB oscillation, magnitude, $M$, appears now in the conductance valley between B3 and B4. This change clearly shows the correlation between those two theoretically expected features and indicates a parity change of the QD state that is moved through the bias window at resonance B3.

## Discussion

Several theories have been striving to explain the experimentally claimed universal occurrence of transmission phase lapses[11–21]. Nonetheless, a satisfactory explanation for such a universal behaviour could not be found. Only for the case where the coupling energy of the QD, $\Gamma$, is larger than the level spacing of the excited states, $\delta$, a universal transmission phase behaviour has been theoretically predicted[17]. In experiment, however, such a regime is difficult to achieve. So far, all experimental works, including ours, have been done in a regime where $\Gamma \sim \delta$ or $\Gamma < \delta$. For these conditions, which are typically encountered in experiment, another set of theory predicts that longer sequences of phase lapses appear for larger QDs, while the probability of finding a phase plateau remains finite[19,21].

In line with previous experimental investigations of a QD having similar dimensions, we find long sequences of phase lapses[6]. Scanning the transmission phase along a larger set of successive resonances, however, we also observe the absence of phase lapses giving rise to phase plateaus. This finding is in

qualitative agreement with the theoretically expected finite probability for phase plateaus of large QDs[21] and clearly shows that the occurrence of phase lapses is not a universal feature as previously claimed. The non-universal transmission phase behaviour is underpinned by the demonstration of altered sequences of phase lapses by QD deformation. The observed tendency of augmented transmission at phase plateaus gives additional support for phase lapse occurrence depending on the parity of the involved QD states[8,9].

In conclusion, our investigations firmly establish that even for the case of a QD containing a few 100 electrons the absence of phase lapses is possible. Our findings show the capability of transmission phase measurements to reveal the microscopic nature of a physical system, such as the orbital parity. We anticipate that the present interferometry experiment opens a path for further studies on other fundamental topics, such as correlated electron systems[41–43].

**Fig. 6** SEM image of the device and scheme of experimental setup. Ohmic contacts are schematically indicated via crossed boxes. The output currents $I_\uparrow$ and $I_\downarrow$ are obtained from the voltages across the resistance, $R = 10$ kΩ

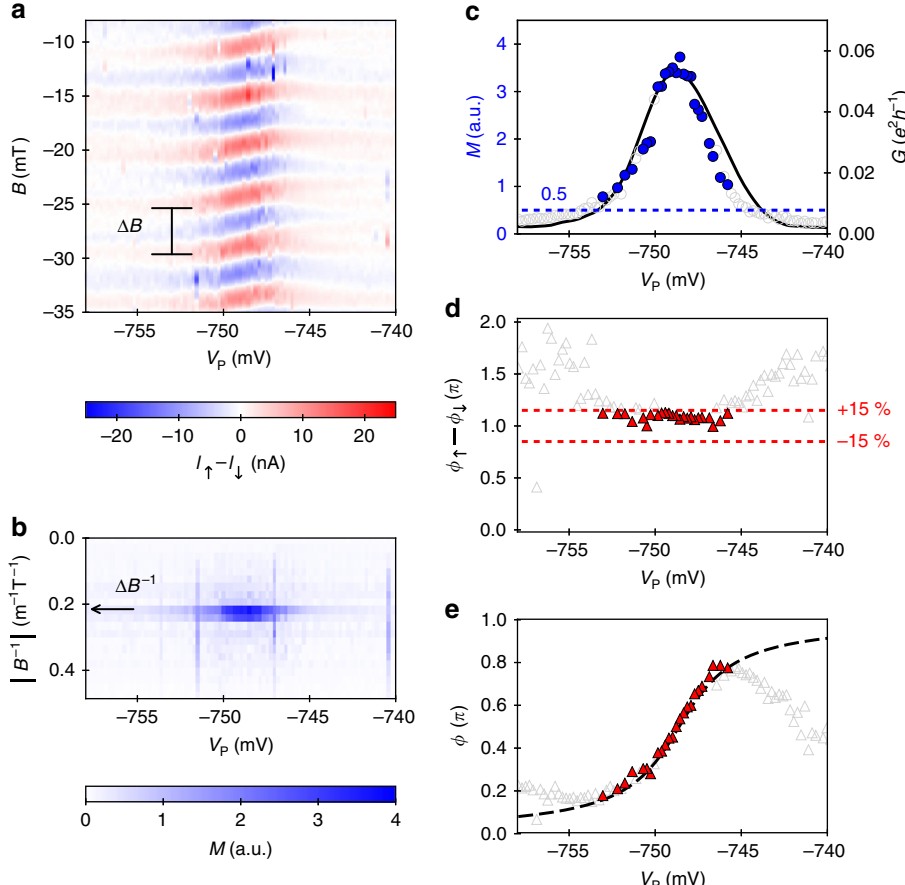

**Fig. 7** Exemplary set of raw data with analysis. **a** Difference of currents, $I_\uparrow - I_\downarrow$, as a function of magnetic flux density, $B$, and the plunger gate voltage, $V_P$. **b** Magnitude of AB oscillation, $M$, obtained by Fourier analysis with respect to $B$. The arrow indicates the periodicity of the AB oscillation, $\Delta B$. **c** Slice of $M$ at $\Delta B$ (left axis; blue points) and electrical conductance through the quantum dot (right axis; black line) as a function of $V_P$. **d** Difference of the AB phases, $\phi_\uparrow - \phi_\downarrow$. **e** Transmission phase, $\phi$, (red points) as a function of $V_P$. The dashed line indicates the evolution expected from a Breit–Wigner profile based on a Lorentzian fit of $G$ see data (**c**). Data points not fulfilling our quality criteria are shown as open, grey symbols

## Methods

**Experimental setup**. Our device is realised with a standard Schottky gate technique in an AlGaAs/GaAs heterostructure hosting a 2DEG with a density, $n \approx 3.2 \cdot 10^{11}$ cm$^{-2}$, and a mobility, $\mu \approx 10^6$ cm$^2$ V$^{-1}$ s$^{-1}$, that is located 110 nm below the surface. The structure of the electron two-path interferometer is formed by Schottky gates, which are realized by electron-beam lithography. A scanning electron microscopy (SEM) image of those electrodes deposited on the surface of the chip is shown in Fig. 6. Applying a set of negative voltages on the electrodes, we locally suppress the 2DEG below and define the conductive paths and the interferometer structure. Electrical connection to the 2DEG is established via ohmic contacts. The AB ring is defined by the central electrode, C, which is connected via a metal bridge over the upper interferometer branch. This electrode allows independent control of the tunnel barriers (TL and TR). The metal bridge is fabricated by two additional electron-beam lithography steps: First a pad of SU-8 photo-resist is deposited. This pad prohibits electrical connection of the metal bridge, which is deposited in the second step. The experiments are performed at a temperature of about 30 mK using a $^3$He/$^4$He dilution refrigerator. To increase the signal-to-noise-ratio, a standard lock-in measurement is performed with a modulation frequency of 23.3 Hz and an amplitude of 20 µV. The output currents, $I_\uparrow$ and $I_\downarrow$, are obtained from the voltages, $V_\uparrow$ and $V_\downarrow$, across a resistance of 10 kΩ that is placed on the chip carrier.

**Data analysis**. The transmission phase measurement is performed by sweeping the magnetic flux density, $B$, and stepping the voltage on the plunger gate, $V_P$. An exemplary set of raw data along a resonance is shown in Fig. 7a. Shown is the difference of the currents, $I_\uparrow$ and $I_\downarrow$, measured at the two terminals at the detection side. From the periodicity of the AB oscillation, $\Delta B = h \cdot e^{-1} \cdot A^{-1} \approx 4.5$ mT, we can deduce the effective area of the AB ring as $A \approx 2.14$ µm × 0.43 µm, which is consistent with the lithographically defined geometry of the Schottky gates. The AB oscillations experience a shift as $V_P$ is scanned along a resonance. This shift directly reflects the course of the transmission phase as a QD state moves through the resonance.

To obtain the transmission phase data, we perform a Fourier transform with respect to $B$. The measured current signals, $I_\uparrow$ and $I_\downarrow$, contain a continuous background. To force the inflexion points of the AB oscillations to zero, we smooth the data and calculate the second derivative. This procedure is comparable to the subtraction of a continuous background and leads to similar results. By forcing the inflexion points to zero, the quality of the Fourier transform is strongly enhanced. Therefore, this approach facilitates the detection of the AB periodicity. Figure 7b shows the magnitude of the Fourier transformed data, $M = |f_\uparrow| + |f_\downarrow|$. The data clearly show a peak (see arrow) corresponding to the AB periodicity, $\Delta B$. For the further analysis, we only process a slice of the data at $B = \Delta B$, which reflects the AB oscillation:

$$f_{\uparrow/\downarrow}^{AB}(V_P) = \mathcal{F}_B\left(\frac{\partial^2 I_{\uparrow/\downarrow}(B, V_P)}{\partial B^2}\right)\Big|_{B^{-1}=\Delta B^{-1}}. \tag{1}$$

In order to perform a reliable phase measurement, two criteria have to be met. The first quality criterion ensures sufficient coherent transmission through the QD and, thus, a sufficient signal-to-noise ratio of the AB oscillations. The second quality criterion assesses the anti-phase relation of the AB oscillations in $I_\uparrow$ and $I_\downarrow$, which assures that multi-path contributions are eliminated[24,25,29,30].

The coherent transmission criterion is implemented via a threshold for the magnitude of AB oscillation, $M$. Figure 7c shows a comparison of a slice of the Fourier transformed data at the AB peak (blue data points) with the electrical conductance through the QD, $G$ (black line). Here $G$ is separately measured by guiding the electrons only through the interferometer branch hosting the QD. $G$ shows the Coulomb blockade peak reflecting the total transmission probability through the QD. The slice of $M$ on the other hand is based on the AB effect—a phase coherent quantum interference phenomena. Therefore, $M$ merely describes the coherent transmission probability and serves as adequate quantity for the assessment criterion. In order to ensure sufficient visibility of the AB oscillations, we set a coherent transmission criterion such that only data points above a threshold of $M > 0.5$ (blue, dashed line) are kept. By this approach, only data points

with a distinctive AB peak in the Fourier transform are kept. AB oscillations with a small signal-to-noise ratio, however, are discarded.

The anti-phase criterion assesses the phase difference of the AB oscillations in the currents, $I_\uparrow$ and $I_\downarrow$. This AB phase can be directly calculated from the argument of the projected Fourier transform:

$$\phi_{\uparrow/\downarrow}(V_P) = \arg\left(f^{AB}_{\uparrow/\downarrow}(V_P)\right). \qquad (2)$$

Figure 7d shows the phase difference of the AB oscillations in the two terminals for the exemplary data set. The quality criterion is set such that only data points are accepted, where the deviation from the anti-phase relation is below the threshold:

$$||\phi_\uparrow - \phi_\downarrow| - \pi| < \pi \cdot 15\%. \qquad (3)$$

Finally, the transmission phase, $\phi$, is extracted only for $V_P$ values where the data fulfil the two quality criteria (filled symbols) by evaluating the expression:

$$\phi = \arg\left(f^{AB}_\uparrow - f^{AB}_\downarrow\right). \qquad (4)$$

The corresponding values of the exemplary data set are shown in Fig. 7e. The red, filled triangles show data points fulfilling our quality criteria. Fitting the conductance, $G$, shown in Fig. 7c with a Lorentzian function, we can construct the expected course of $\phi$ from Breit–Wigner resonance theory (dashed line). The data points fulfilling our quality criteria agree with the expected course, whereas the discarded data points (open triangles) strongly deviate.

**Data availability**. The data that support the findings of this study are available from the corresponding authors on reasonable request.

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

## Acknowledgements

We acknowledge fruitful discussions with Rodolfo Jalabert and Dietmar Weinmann. Additionally, we thank Xuedong Hu for critical reading of the manuscript and Dominique Mailly for teaching us the employed bridge technique. Sh.T. acknowledges financial support from the European Union's Horizon 2020 research and innovation program under the Marie Skłodowska-Curie grant agreement No. 654603. M.Y. acknowledges financial support by PRESTO, JST (No. JPMJPR132D) and Grant-in-Aid for Scientific Research A (No. 26247050). Se.T. acknowledges financial support by JSPS, Grant-in-Aid for Scientific Research S (No. 26220710), CREST (JPMJCR15N2, JPMJCR1675) and QPEC, the University of Tokyo. A.L. and A.D.W. acknowledge gratefully support of DFG-TRR160, BMBF - Q.com-H 16KIS0109, and the DFH/UFA CDFA-05-06. C.B. acknowledges financial support from the French National Agency (ANR) in the frame of its program BLANC FLYELEC Project No. anr-12BS10-001. This project has received funding from European Union's Horizon 2020 research and innovation program under the Marie Skłodowska-Curie grant agreement No. 642688.

## Author contributions

Sh.T., M.Y., Se.T., T.M. and C.B. initiated this work. H.E. and Sh.T. did the experiment, analysed the data and wrote the manuscript with assistance from T.M. and C.B. Sh.T. fabricated the sample, G.R. contributed to the experimental set up. A.L. and A.D.W.

provided the GaAs heterostructure. All authors discussed the results and commented on the manuscript. C.B. supervised the work.

## Additional information

**Competing interests:** The authors declare no competing financial interests.

