## [Peer Review File · Nature Communications]

Reviewers' Comments:

Reviewer #1:

Remarks to the Author:

The authors present a very intriguing study on electronic wave functions in a Mach-Zehnder-type configuration including a large quantum dot. These experiments are performed in the spirit of the so-called 'Which-path-detectors' exhibiting transmission phase jumps of the electronic wave functions.

The authors took great care in preparing their experiment and performed it in a very defined fashion. The interferometer as well as the quantum dot are well characterized and the measurements appear to have been conducted with great care. The major finding of the authors is that -- in the limit of coupling energy equalling level spacing -- the transmission phase is non-universal.

While the major findings are very clearly represented and worth publishing I am only wondering about the overall aim of the work, i.e. is the claim that transmission phases are truly non-universal or only for their specific 200-electron quantum dot. I am asking this, since the discussion of wave-function parity is left a bit open ended and so is the overall 'non-universality'. The authors need to clarify this point before publication. Also, it appears to me that the true quest is for level spacing exceeding the coupling energy, but maybe this is technically too difficult.

Reviewer #2:

Remarks to the Author:

The authors measure the evolution of the transmission phase through a large quantum dot under variation of the gate voltage over a range that covers 14 resonances. They find for most of the Coulomb-blockade valleys between consecutive resonances a phase lapse, i.e. a jump from π down to 0 (universal behavior), but in some rare cases an absence of such a phase lapse (non-universal). With their experiment, the authors aim at shedding light on "the longest standing puzzles of mesoscopic physics", namely an "explanation for the complete absence of TP [transmission phase] plateaus and the question about a universal TP behavior".

In my opinion, is written in a very clear way, the addressed problem is clearly stated and nicely embedded in the existing literature. I am impressed by the quality of the data, which is achieved by a clever sample design and a convincing data analysis. This improved data quality, as compared to previous experiments, is the most important result of the manuscript in the present form. There is no doubt that the reported experiment defines an advancement that warrants publication in some form.

Therefore, the remaining question is whether or not the reported findings justify publication in Nature Communications. For me, this is linked to the question of whether and to what extent "the longest standing puzzles of mesoscopic physics" can be solved. So, what do I learn about the solution of this puzzle after reading the manuscript? I am afraid, the answer is: not much. Yes, now there is a new experiment in the regime of a large quantum dot that shows some deviations from the universal behavior. But what does this imply for the theoretical understanding? The theoretical prediction of the universal behavior is mainly based on the assumption that there is one level that is much stronger coupled to the leads than the other levels such that, effectively, it is always the same level that is being occupied with increasing gate voltage over and over again. What do the authors conclude from their measurements: is this picture valid or not?

The quality of the experiment such so good that I strongly believe the authors can deduce more information that might be helpful to contribute to solving the puzzle. It would be nice to have a

more specific understanding of the dot configuration in each of the Coulomb-blockade valleys. Two ideas come to my mind:

1. The authors deduce an average level spacing. Is there any possibility to access the excitation spectrum from the dI/dV measurements to get the individual level positions? And does this, then, help to understand the positions at which the plateaus appear.

2. How about the spin degree of freedom? The authors state that they do not see Kondo features and they relate this to the fact that the level spacing is small such that there are several levels participating. As additional pieces of information I would like to see an estimate of the Zeeman splitting as compared to the other system parameters. Maybe, one can also give a number for the Kondo temperature (If only a single level was involved).

But not only in the Kondo regime, also for weaker coupling, spin is important in quantum-dot Aharonov-Bohm interferometry. As proposed in PRL 86, 3855 (2001) and PRB 65, 045316 (2002) and experimentally confirmed in PRL 92, 176802 (2004) and NJP 9, 111 (2007), the amplitude of the Aharonov-Bohm oscillation should depend on the spin state of the quantum dot: if the participating dot level is predominantly empty then the oscillation amplitude should be larger than in the case when the spin-degenerate level is singly occupied since spin-flip processes during the transmission through the dot reduce the interference. In the present experiment, the resonance is clearly marked by the phase $\pi/2$. When going to the left and right of this resonance: does the Aharonov-Bohm oscillation amplitude show the same or a different reduction (the described spin physics yields the second behavior)? Looking at Fig. 4 with my naked eye, I have the feeling that such asymmetries might be there and they might be systematically linked to the appearance of the TP plateaus. If this was the case, then the present experiment would definitely deliver new insight on the puzzle that was not available in previous experiments.

In conclusion, I recommend to further analyze the data and, if new insight comes up, revise the manuscript accordingly.

Response to the comments of the reviewers

Reviewer #1 (Remarks to the Author):

The authors present a very intriguing study on electronic wave functions in a Mach-Zehnder-type configuration including a large quantum dot. These experiments are performed in the spirit of the so-called 'Which-path-detectors' exhibiting transmission phase jumps of the electronic wave functions.

The authors took great care in preparing their experiment and performed it in a very defined fashion. The interferometer as well as the quantum dot are well characterized and the measurements appear to have been conducted with great care. The major finding of the authors is that -- in the limit of coupling energy equalling level spacing -- the transmission phase is non-universal.

While the major findings are very clearly represented and worth publishing I am only wondering about the overall aim of the work, i.e. is the claim that transmission phases are truly non-universal or only for their specific 200-electron quantum dot. I am asking this, since the discussion of wave-function parity is left a bit open ended and so is the overall 'non-universality'. The authors need to clarify this point before publication. Also, it appears to me that the true quest is for level spacing exceeding the coupling energy, but maybe this is technically too difficult.

1.1. Response to Reviewer #1:

The overall aim of our work is to show that the transmission phase (TP) depends on the subtle internal structure of the quantum dot (QD) (i.e. the wave functions of the QD states), which results in a *truly* non-universal TP behaviour. Our observations of TP plateaus and the alternation of the TP lapse sequence via QD deformation clearly show that the TP is non-universal for QDs of comparable, smaller and presumably even larger size [R2]. In addition, we demonstrate that QD deformation alters the sequence of TP lapses and plateaus. This new finding shows that even for a large QD the occurrence of TP lapses is not independent of the QD shape and clears up previous (wrong) claims. Our findings close the debate about a universal TP behaviour that arose from the results of the pioneering experiment of Schuster et al. [R1]. In addition, our experimental data gives much deeper insight into the TP problem than any other preceding experimental work. The characteristic features that are observable in our TP data will provide important contributions for a better understanding of the TP problem.

Another important new feature of the present data is that the augmented AB oscillation amplitude in conductance valleys is *directly linked* to the occurrence of TP plateaus. Such a characteristic is theoretically predicted in particular for a parity-dependent TP behaviour [R4,R5]. This feature, which we observe for the first time in experiment, helps to strengthen the theoretical understanding of the TP problem. The correspondence of the TP lapse sequence to parity changes of the QD states in addition demonstrates the potential of TP measurements to access the internal structure of QDs.

The theoretical work of Karrasch et al. [R7] that we mention in the discussion section of our manuscript indeed suggests that for a particular regime ($\Gamma > \delta$) a universal TP behaviour might occur. Let us emphasize that so far **all previous experimental works were carried out in the $\Gamma \leq \delta$ regime**. This is true for small few electron QDs as well for large QDs containing several hundred

electrons (i.e. our QD). Whether a **universal regime** will appear for even larger QDs, consequently, **is absolutely hypothetical**. Nonetheless, it is indeed an interesting – though, also a very challenging – question if such a regime can be experimentally achieved. One could try to investigate an extremely large QD. For such a situation, however, due to the very small charging energy, one will have to cope with experimental limitations coming for instance from gate voltage fluctuations. Let us remark at this point that another theoretical work of Molina et al. [R2] suggests that even for much larger QDs such phase plateaus appear, however with reduced probability. According to this theory only for the **hypothetical case of an infinitely large** QD only TP lapses would occur.

Let us emphasize that the aim of our work is to solve the puzzle about a universal TP behaviour that arose from previous experiments performed in the $\Gamma \leq \delta$ regime. The investigation of the $\Gamma > \delta$ regime is not the scope of our present work. By highlighting this issue in the discussion section of our manuscript, nonetheless, we anticipate that our work will stimulate further experimental efforts towards that direction.

We have revised our manuscript in order to make the overall-aim and the impact of our work clearer. In particular we clarified to what extent one can access the parity of the QD states via TP measurements. We now emphasize more explicitly the clear correspondence between the finite coherent transmission in the conductance valleys and TP plateau occurrence. In addition, we highlight that this feature is indeed reminiscent for two adjacent orbital levels having opposite parity [R4,R5].

Reviewer #2

The authors measure the evolution of the transmission phase through a large quantum dot under variation of the gate voltage over a range that covers 14 resonances. They find for most of the Coulomb-blockade valleys between consecutive resonances a phase lapse, i.e. a jump from π down to 0 (universal behavior), but in some rare cases an absence of such a phase lapse (non-universal). With their experiment, the authors aim at shedding light on "the longest standing puzzles of mesoscopic physics", namely an "explanation for the complete absence of TP [transmission phase] plateaus and the question about a universal TP behavior".

In my opinion, is written in a very clear way, the addressed problem is clearly stated and nicely embedded in the existing literature. I am impressed by the quality of the data, which is achieved by a clever sample design and a convincing data analysis. This improved data quality, as compared to previous experiments, is the most important result of the manuscript in the present form. There is no doubt that the reported experiment defines an advancement that warrants publication in some form.

Therefore, the remaining question is whether or not the reported findings justify publication in Nature Communications. For me, this is linked to the question of whether and to what extend "the longest standing puzzles of mesoscopic physics" can be solved. So, what do I learn about the solution of this puzzle after reading the manuscript? I am afraid, the answer is: not much. Yes, now there is a new experiment in the regime of a large quantum dot that shows some deviations from the universal behavior. But what does this imply for the theoretical understanding?

2.1. Response to Reviewer #2:

First, let us shortly introduce the long-standing puzzle we are talking about:

The puzzle about a universal transmission phase behaviour arose about 20 years ago from the interpretation of important experimental observations in the pioneering studies on transmission phase (TP) behaviour [R1]: In between all of the *investigated* resonances, TP lapses were observed. It was claimed that a *universal* TP behaviour was observed, where *only* TP lapses occur. The result led to a lot of theoretical effort trying to explain this surprising universal feature. For the situation investigated by Schuster et al. – a large quantum dot (QD) hosting ≈ 200 electrons with $\Gamma < \delta$ –, however, so far no sufficient explanation for such a universal regime was found. In order to solve this puzzle, we conducted TP measurements for a larger QD similarly in the $\Gamma \leq \delta$ regime with an original effective three-terminal two-path interferometer allowing very reliable TP measurements.

So what can one learn from our manuscript about this puzzle and what does it imply for the theoretical understanding?

(i) TP plateaus occur even for a large QD.

Let us emphasize that up to now **all experimental works** have been performed on QDs ranging from few electron QDs up to QDs containing several hundred electrons (i.e. our QD) and that **for all these QDs** the regime $\Gamma \leq \delta$ applies. Investigating a larger parameter space with a novel electron two-path interferometer, we reproduce the original observations of several consecutive TP phase lapses. Besides those successive sequences, however, we observe **for the first time also absences of TP lapses in a large QD**. This *significant* deviation from the universal TP behaviour puts the observations of the pioneering experiment [R1] in a new perspective: It shows that there

is so far not a single experiment which could show that the TP evolution is universal. Our experiment therefore solves the long standing puzzle about the universal TP behaviour claimed for a large QD in the $\Gamma \leq \delta$ regime.

(ii) The QD size determines the probability of TP lapse absence.

Our observation of TP plateaus is in line with the theoretical investigation of Ref. [R2], where it was shown that the probability for TP lapse absence should decrease with QD size. Such a relation explains the occurrence of long sequences of successive TP lapses in a large QD, but answers the question about a universal feature negatively. In addition, there were theoretical efforts [R2,R3] to calculate the probability of TP lapse absence. Our TP data along 14 successive resonances provides data giving an idea about the frequency of TP plateau occurrence in a QD hosting about 300 electrons ($\approx 20\%$). For such theoretical works, therefore, our TP data serves as important reference. We anticipate that our work will stimulate further theoretical and experimental efforts to quantify the probability of TP lapse absence depending on the QD properties.

(iii) There are features of parity-dependent TP lapse occurrence.

Early theoretical investigations [R4,R5] proved that the sequence of the orbital parities successively crossing the bias window is the decisive factor for the TP lapse sequence. Orbitals having same parity lead to a transmission zero in the interjacent conductance valley, which gives rise to a TP lapse. For QD orbitals having opposite parity, on the other hand, such a transmission zero is absent. This results in a finite *coherent* transmission in the conductance valley and a cumulated TP: a TP plateau. We indeed observe the feature of augmented transmission in conductance valleys where TP lapses are absent. Our observation is in congruency with the aforementioned theoretical expectation [R4,R5] and it demonstrates the potential of TP measurements to access the parity of the QD orbitals.

(iv) The sequence of TP lapses is *truly* non-universal also in a large QD.

In previous works it was claimed that for a large QD the sequence of TP lapses is independent of the QD shape [R1,R6]. In our work we demonstrate the opposite: As we sufficiently deform the QD, the sequence of TP lapses and plateaus changes. This finding indicates that the internal structure of the QD indeed affects the sequence of TP lapses and plateaus.

Reviewer #2

The theoretical prediction of the universal behavior is mainly based on the assumption that there is one level that is much stronger coupled to the leads than the other levels such that, effectively, it is always the same level that is being occupied with increasing gate voltage over and over again. What do the authors conclude from their measurements: is this picture valid or not?

2.2. Response to Reviewer #2:

The theoretical work of Karrasch et al. [R7] that we mention in the discussion section of our manuscript indeed suggests that for a particular regime ($\Gamma > \delta$) a universal TP regime might occur. Let us emphasize that so far **all previous experimental works were carried out in the $\Gamma \leq \delta$ regime**. This is true for small few electron QDs as well for large QDs containing several hundred electrons (i.e. our QD). Whether a **universal regime** will appear for even larger QDs, consequently, is **absolutely hypothetical**. Nonetheless, it is indeed an interesting – though, also a very challenging – question if such a regime can be achieved in experiment. One could try to investigate an extremely large QD. For such a situation, however, due to the very small charging energy, one will

have to cope with experimental limitations coming for instance from gate voltage fluctuations. Let us remark at this point that another theoretical work [R2], suggests that even for larger QDs such phase plateaus appear, however with reduced probability. According to this theory only for the **hypothetical case of an infinitely large QD** only TP lapses would occur.

Let us emphasize that the aim of our work is to solve the puzzle about a universal TP behaviour that arose from previous experiments performed in the $\Gamma \leq \delta$ regime. The investigation of the $\Gamma > \delta$ regime is not the scope of our present work. By highlighting this issue in the discussion section of our manuscript, nonetheless, we anticipate that our work will stimulate further experimental efforts towards that direction.

Reviewer #2 (Remarks to the Author):

The quality of the experiment such so good that I strongly believe the authors can deduce more information that might be helpful to contribute to solving the puzzle. It would be nice to have a more specific understanding of the dot configuration in each of the Coulomb-blockade valleys. Two ideas come to my mind:

1. The authors deduce an average level spacing. Is there any possibility to access the excitation spectrum from the dI/dV measurements to get the individual level positions? And does this, then, help to understand the positions at which the plateaus appear.

2.3. Response to Reviewer #2:

In order to measure a detailed excitation spectrum, one has to make the coupling energy, Γ , small enough compared to the level spacing of the excited states, δ , while keeping the symmetry of the coupling to the left and the right lead to ensure large enough conductance through the QD. Since Γ is exponentially dependent on the thickness of the tunnel-barrier it is difficult to keep the symmetry while keeping Γ small. In addition the symmetry of the tunnel-barrier also changes slightly due to the plunger gate voltage sweep during the transconductance measurement. As a result, it is difficult to obtain a detailed excitation spectrum over the whole range of Coulomb-blockade peaks (CPs) in such a gate-defined lateral QD. Despite those limitations we were able to observe features of the excitation spectrum for a few specific resonances that enable us to roughly estimate the magnitude of δ . This approach is valid since one expects that δ hardly changes when several electrons are loaded into a large QD containing already a few hundred electrons.

From the sequence of TP lapses and plateaus, we can deduce the sequence of parity changes. In between the three TP plateaus, we observe respectively five and two successive TP lapses. We also tried to obtain information about the spin of the QD states (see response 2.5). Our measurements show that the internal structure of the QD is quite complex: The situation is different from the naively expected situation, where a quantised energy level of a QD is filled up with two electrons having opposite spins one after another from lower to higher energy. Our data rather indicates that for a large QD electrons are loaded one after another in different orbitals as the plunger gate voltage moves the energy level through the bias window. Due to the complex structure of the present QD hosting hundreds of electrons, we suppose that even with the information provided by a detailed excitation spectrum one can hardly model the occurrence of TP plateaus.

Reviewer #2 (Remarks to the Author):

How about the spin degree of freedom? The authors state that they do not see Kondo features and they relate this to the fact that the level spacing is small such that there are several levels participating. As additional pieces of information I would like to see an estimate of the Zeeman splitting as compared to the other system parameters. Maybe, one can also give a number for the Kondo temperature (if only a single level was involved).

2.4. Response to Reviewer #2:

We disagree with comment of reviewer #2: "... the authors state that they do not see Kondo features and they relate this to the fact that the level spacing is small such that there are several levels participating."

We do not state that. We only mention a general statement in the introduction section about the expected phase shift in the Kondo regime [R8], which occurs in the strong coupling regime (large Γ). In our present experiment **we are rather in the weak coupling regime (small Γ)**. We reduce the coupling in order to suppress the conductance in the Coulomb blockade valley such that the Kondo temperature is lower than the electron temperature. As a result, we do not observe any signature of the Kondo effect. To make this clearer, we have rearranged the corresponding paragraph in the introduction section.

Concerning the question about the Zeeman splitting and the Kondo temperature:

For the investigated magnetic field range the Zeeman splitting is smaller than temperature fluctuation – the minimum energy scale in our experiment. Thus, it is also smaller than the charging energy, the coupling energy and the single-level spacing. We did not observe a conductance enhancement in any of the investigated valleys with temperature. We hence conclude that the Kondo temperature is lower than the electron temperature. We added corresponding comments about those energy scales to the manuscript.

Reviewer #2 (Remarks to the Author):

But not only in the Kondo regime, also for weaker coupling, spin is important in quantum-dot Aharonov-Bohm interferometry. As proposed in PRL 86, 3855 (2001) and PRB 65, 045316 (2002) and experimentally confirmed in PRL 92, 176802 (2004) and NJP 9, 111 (2007), the amplitude of the Aharonov-Bohm oscillation should depend on the spin state of the quantum dot: if the participating dot level is predominantly empty then the oscillation amplitude should be larger than in the case when the spin-degenerate level is singly occupied since spin-flip processes during the transmission through the dot reduce the interference. In the present experiment, the resonance is clearly marked by the phase $\pi/2$. When going to the left and right of this resonance: does the Aharonov-Bohm oscillation amplitude show the same or a different reduction (the described spin physics yields the second behavior)? Looking at Fig. 4 with my naked eye, I have the feeling that such asymmetries might be there and they might be systematically linked to the appearance of the TP plateaus. If this was the case, then the present experiment would definitely deliver new insight on the puzzle that was not available in previous experiments.

In conclusion, I recommend to further analyze the data and, if new insight comes up, revise the manuscript accordingly.

2.5. Response to Reviewer #2:

We also considered that spin-flip processes play a role in the transmission through the QD and that they affect the shape of the peaks of the AB oscillation amplitude (OA).

When comparing the OA and conductance peaks, we find interesting features:

i) In the valley without a TP lapse, the OA is enhanced compared to the other valleys, whereas such a feature does not appear in conductance. This feature is present whenever we see a transmission phase plateau. As discussed in the manuscript we understand this feature in terms of an absent transmission zero that occurs when the orbital parity is changing.

ii) We also observe an asymmetric shape of the OA peaks. This feature can come from three effects –the sequence of transmission zeros [R4,R5], spin-flip processes [R9-14] or the Fano-effect [R15-17] –, which are difficult to distinguish.

We could not find any systematic dependence about the appearance of such an asymmetry. Therefore, we did not discuss this feature in the first version of the manuscript.

We agree with reviewer #2 that a comparison of the AB oscillation to the conductance peaks gives more insight into the system. In addition it shows the complexity of the electron filling procedure for a large QD and might be helpful to contribute to a better understanding of the TP. Hence, we have added conductance data to Fig. 4 and 5 in addition to the OA with a discussion for the reader including the references that were pointed out by the reviewer and a comparison of the OA and conductance data.

As a final remark we would like to emphasize that TP measurements are very difficult to realize and only a few experiments are available. In our manuscript we present high quality data, which will serve as important reference. Our observations clearly show a non-universal TP behaviour for a large QD. This finding solves the puzzle about the universal TP behaviour claimed from Ref. 1 and, therefore, sets a landmark contribution for mesoscopic physics.

References:

- [R1] Schuster, R. *et al. Nature* **385**, 417-420 (1997)
- [R2] Molina, R. A. *et al. Phys. Rev. Lett.* **108**, 076803 (2012)
- [R3] Jalabert, R. A. *et al. Phys. Rev. E* **89**, 052911 (2014)
- [R4] Lee, H.-W. *Phys. Rev. Lett.* **82**, 2358 (1999)
- [R5] Taniguchi, T. & Büttiker, M. *Phys. Rev. B* **60**, 13814 (1999)
- [R6] Avinun-Kalish, M. *et al. Nature* **436**, 529-533 (2007)
- [R7] Karrasch, C. *et al. Phys Rev. Lett.* **98**, 186802 (2007)
- [R8] Takada, S. *et al. Phys Rev. Lett.* **113**, 126601 (2014)
- [R9] Aker, H. *et al. Phys. Rev. B* **47**, 6835 (1993)
- [R10] Aker, H. *et al. Phys. Rev. B* **59**, 9802 (1999)
- [R11] König, J. & Gefen, Y. *Phys Rev. Lett.* **86**, 3855 (2001)
- [R12] König, J. & Gefen, Y. *Phys Rev. B* **65**, 045316 (2002)[R13] Aikawa *et al. Phys Rev. Lett.* **92**, 176802 (2004)
- [R14] Ihn, T. *et al. New J. Phys.* **9**, 111 (2007)
- [R15] Kobayashi, K. *et al. Phys Rev. Lett.* **88**, 256806 (2002)
- [R16] Kobayashi, K. *et al. Phys Rev. B* **70**, 035319 (2004)
- [R17] Aikawa, H. *et al. J. Phys. Soc. Jpn.* **73**, 3235 (2004)

In the following we provide a list of major modifications, where paragraphs are rewritten or new contents are added. By revising the manuscript we also corrected typos and enhanced the wording in several sentences. These minor modifications are not listed in the following. Nonetheless, all (major and minor) modifications are highlighted in **green** in the revised version of the manuscript.

List of major modifications:

- Page 2; Section: “Introduction”:
 - Lines 30-41:
Rephrased paragraph for making a better transition to the central question.
- Page 6; Section: “Properties of the quantum dot”:
 - Lines 139-140:
Comparison of the Zeeman energy with typical energy scales of the experiment.
 - Lines 140-142:
Comment on Kondo temperature.
- Page 8-10; Section: “Transmission phase measurements”:
 - Lines 154-163:
More detailed and clearer description of TP measurements.
 - Lines 164-181:
Detailed description of transmission zero and connection with augmented AB oscillation amplitudes in conductance valleys with TP plateau.
 - Lines 182-189:
Fig. 4 contains now additional conductance data for comparison with the AB oscillation amplitude.
 - Lines 190-204:
Discussion of asymmetry in AB oscillation amplitude and spin-flip processes including eight new references (R9-16).
- Page 10-11; Section: “Modifying the quantum dot shape”:
 - Line 216-223:
Rephrased discussion with highlighted relation to the parity of the QD states.
 - Line 224-231:
Also Fig. 5 contains now additional conductance data for comparison with the AB oscillation amplitude. The arrows now additionally indicate the QD-deformation introduced changes on the TP.

Reviewers' Comments:

Reviewer #1:

None

Reviewer #2:

Remarks to the Author:

The authors have spend great effort to address all the points raised by both reviewers. The manuscript is now much clearer, in particular in pointing out the differences to earlier works. I am now convinced that the manuscript warrants publication in Nature Communications.